# The Role of Science-Based Knowledge on the SARS-CoV-2 Virus in Reducing COVID-19-Induced Anxiety among Nurses

**DOI:** 10.3390/ijerph19127070

**Published:** 2022-06-09

**Authors:** Ilana Dubovi, Angela Ruban, Anat Amit Aharon

**Affiliations:** 1Nursing Department, Sackler Faculty of Medicine, Tel Aviv University, Tel Aviv 6997801, Israel; angellruban@tauex.tau.ac.il (A.R.); anatamit@tauex.tau.ac.il (A.A.A.); 2Sackler Faculty of Medicine, Sagol School of Neuroscience, Tel Aviv University, Tel Aviv 6997801, Israel

**Keywords:** nurses, nursing, COVID-19, anxiety, knowledge, science-based knowledge

## Abstract

The COVID-19 infection has generated not only a risk of morbidity and mortality but also resulted in an enormous psychological impact on healthcare providers and the general public. This study aimed to evaluate the prevalence of anxiety and identify the role of protective factors. A two-part cross-sectional study was conducted, by means of an online questionnaire. Part 1 investigated 562 registered nurses, nursing students, and the general public. Participants were assessed for anxiety symptoms with the State-Trait-Anxiety Inventory. A one-way ANCOVA analysis revealed that nurses had the highest level of anxiety compared to the general public and students, with 26% of them reporting severe anxiety. To identify how anxiety can be mitigated, the Part 2 study was focused on registered nurses from Part 1. Multiple regression revealed that a higher level of science-based knowledge of COVID-19 and professional experience were associated with a lower level of anxiety among nurses. The findings suggest that nurses are a vulnerable population prone to anxiety symptoms resulting from the COVID-19 pandemic. Having a deeper science-based understanding of COVID-19 may protect nurses from anxiety. This study underlines the importance of deep evidence-based knowledge for health providers, which may be generalized to a possible future emergency disaster.

## 1. Introduction

The global outbreak of the novel COVID-19 pandemic has imposed enormous pressure on governments, medical and healthcare providers, and the general public. The COVID-19 viral infection has brought not only the risk of death but also reactionary strict isolation measures, which have resulted in social distancing, school closures, and income instability. In Israel, as of mid-May 2020, there were 16,689 confirmed cases of COVID-19, with 266 deaths (a prevalence of 29/1 million citizens). In late April and early May, Israel was ranked 47th in the number of deaths per one million citizens and 25th in the number of confirmed cases per one million citizens relative to 210 countries worldwide [1].

Building upon previous studies from the SARS and Ebola epidemics and from data on the COVID-19 infection outbreak, it is reasonable to expect a consequential emotional impact on individuals in terms of increased anxiety [2,3,4]. While a low level of anxiety is helpful to motivate and generate excitement in an individual, high levels of anxiety may become excessive and interfere with daily life and lead to negative consequences [5,6]. Indeed, studies have highlighted the negative effects of higher levels of anxiety regarding COVID-19 on psychophysiological health. This includes a reduced appetite, dizziness, sleep disturbance, and nausea or vomiting [7]. Such COVID-19-induced anxiety may also lead to reduced work performance and job satisfaction, which is associated with frequent absenteeism and eventual job loss [2,8]. A high level of pandemic anxiety was associated with higher burnout, depression, and fear among nurses and other healthcare workers [9,10]. Pandemic-related stress has also been linked with cognitive functional impairment, such as reduced attention and working memory; negative coping mechanisms, such as increased intake of alcohol or drugs; and increased suicidal ideation [11]. Spielberger et al. differentiated between anxiety as a temporary emotional “state” and as a consistent personality attribute or “trait” [12]. The current study focuses on state anxiety, which refers to a condition of feeling fearful, tense, apprehensive, nervous, and worried due to physiological arousal or a threatening stimulus. State anxiety refers to a specific threat of a shorter duration that may disappear as the threat weakens [13]; in this case, the COVID-19 outbreak [14]. Spielberger also developed the State-Trait Anxiety Inventory (STAI) to assess anxiety [12], which has been validated among students, adults, medical patients, health care workers, and recently in the context of the COVID-19 pandemic [15,16,17].

Though the COVID-19 pandemic has psychologically affected most people worldwide, various population groups have been reported to be particularly vulnerable [18]. Studies show that nurses, as frontline responders to the COVID-19 pandemic, have been reported to experience the highest prevalence and levels of anxiety [19,20,21]. Working on the frontline involved wearing personal protective equipment and continuous fear of self-infection or of infecting someone in their family. Given that this can trigger anxiety, recent studies pointed out that working with COVID patients was associated with traumatic stress and COVID stress syndrome [22,23]. A systematic review and meta-analysis by Pappa et al. (2020) concluded that the pooled prevalence of anxiety among healthcare workers during the COVID-19 pandemic was 23.21% (95% CI 17.77–29.13, I^2^ = 99%), and among nurses and physicians, 17.93% suffered from mild anxiety. However, the research in this area is inconsistent; for instance, a recent meta-analysis reported that the prevalence of anxiety was similar among nurses and the general public [18,24]. Moreover, other vulnerable groups were identified: college students who are experiencing a fragile transitional phase toward independence and adulthood were more likely to show anxiety symptoms [25]. Students’ increased COVID-19-induced psychological pressure may be explained by various factors. These include family income instability, which can affect the ability to pay tuition fees; academic delays; new remote study methods; travel restrictions; and being confined away from home [26]. Although the entire population is affected by the COVID-19 pandemic, specific segments of the population are experiencing it differently, requiring identification of high-risk groups to set intervention priorities.

Furthermore, building upon the recent call by the Lancet Psychiatry [27] for immediate research into COVID-19-induced anxiety reduction interventions, the current study aims to identify anxiety protective factors. Limited knowledge about COVID-19 was recently pointed out as a barrier and trigger for psychological stress [28]. Evidence in cognitive science suggests that there are two levels of information processing that warrant consideration: surface and deep [29]. Surface processing is the learning of factual or procedural information that does not require one to think about the meaning behind the concept. Conversely, deep processing is associated with learning that allows one to meaningfully interpret information, obtaining an understanding of causal links regarding how and why certain things work [30,31,32,33,34]. Having more cohesive causal information promotes better retrieval of previously learned facts and allows people to make predictions, understand implications, draw inferences, and offer explanations, all of which are necessary for problem-solving protective behaviors during an infectious disease outbreak [35,36,37].

The research aim was twofold: first, to evaluate the prevalence of state anxiety distress among nurses compared to other population groups; second, to investigate the association between factual–procedural and deep science-based information and anxiety levels among professional registered nurses. In order to fulfill these aims, a two-part study was conducted.

## 2. Methods

### 2.1. Research Design and Participants

A two-part, cross-sectional study was conducted in Israel during mid-May 2020, using a survey instrument. The period of the study was during the last stage of the first quarantine in Israel and at the beginning of lockdown easing and reopening of local businesses. The sample in Part 1 of the study included registered nurses, sophomore nursing students, and lay adult individuals from the general public who were not working in healthcare settings. Part 2 of the study included registered nurses who had participated in Part 1.

Sophomore nursing students were recruited via the Nursing Department at one of the largest universities in Israel. We recruited from the second academic year pool only due to the university’s IRB regulations. The students studying in the university’s Department of Nursing come from all regions of the country, Jews and Arabs, and secular and religious. The sampling method was based on a convenience sample.

Registered nurses and individuals from the general public were recruited through an online polling service (*iPanel*, https://www.ipanel.co.il/en/ (accessed on 10 May 2020)) that enables rapid attainment of responses with representative sampling by socioeconomic status, gender, age, and profession. This is the largest panel survey platform in Israel, and it adheres to the high-quality research code of the European Society for Opinion and Marketing Research (ESOMAR) [38]. Although the participants were recruited from all over the country, the sampling method was based on a convenience sample.

The study was conducted after receiving the approval of the Ethics Committee of the University (#1444-1). Before the online questionnaires were administered, participants were given information on the study and were asked to sign an informed consent form, while the computer application maintained the anonymity of the respondents.

### 2.2. Data Collection Instruments

#### 2.2.1. Part 1

The State-Trait Anxiety Inventory—State Anxiety (STAI-S) scale is a widely used instrument composed of 20 self-rated items. It requires participants to rate their subjective experienced intensity of each described feeling (1 = not at all, 4 = very much so) at the moment of assessment [12,39]. The total score ranges from 20 to 80 points, with a higher score indicating higher anxiety levels. The STAI-S is an appropriate measurement of COVID-19-pandemic-induced anxiety since it was designed to measure a temporary emotional state induced by external or internal stimuli [40]. A cut-off score of 40 has been suggested to detect clinically significant symptoms, and a cut-off score higher than 55 denotes severe anxiety [41,42]. In the current study, the internal consistency measured using Cronbach’s alpha coefficient was 0.94, similar to previous reports.

Demographic characteristics. These included gender, age, health status, family income level, and education level. Registered nurses were also asked in which departments and clinics they were working and about their professional seniority level.

#### 2.2.2. Part 2

In Part 2 of the study, nurses who had participated in Part 1 were asked to complete a multiple-choice questionnaire developed by the authors [37], to assess their knowledge of COVID-19. The first part of the questionnaire included three items that evaluate deep *science-based* knowledge of COVID-19 related to transmission modes, causes, and possible treatments. For example, “You read that scientists made a breakthrough on coronavirus antibody injections. What does this breakthrough mean?” Response options are: “a specific treatment for COVID-19”, “active immunity”, “a vaccine to induce the body to produce its own antibodies”, or “a vaccine to produce herd immunity”. Cronbach’s alpha yielded a good internal consistency score of 0.74.

The second part of the questionnaire includes four items that assess factual *procedural* or “how-to” knowledge, which incorporates the necessary behaviors and required competencies to prevent COVID-19 transmission. For example, “Which of the following can protect you from being infected by COVID-19?” Response options are: “disinfecting skin with bleach”, “drinking alcohol”, “drinking hot lemon juice”, “exposure to direct sunlight”, “washing hands with running water and soap”, “taking prescribed antibiotics”, “taking non-steroidal anti-inflammatory drugs (NSAIDs) such as Ibuprofen”, “smoking cigarettes”, and “maintaining a physical distance of about six feet from another person”. Cronbach’s alpha was 0.72.

### 2.3. Statistical Analysis

Descriptive statistics were used to describe the participants’ demographic characteristics, anxiety, and knowledge levels. A one-way ANOVA and a one-way ANCOVA were conducted to examine differences in sociodemographic characteristics and anxiety levels between nurses, students, and general public groups, and to detect the differences between nurses’ work departments on anxiety. To understand the impact of the nurses’ characteristics and both types of knowledge on the variability of COVID-19-induced anxiety, we conducted a hierarchical multiple regression. Preliminary analyses were conducted to ensure that there were no violations of the assumptions of normality, linearity, and homoscedasticity. The order of the explanation variables was as follows: In the first step, the socioeconomic variables were entered to control for their possible effects; in the second step, nurses’ working variables were entered; and in the final step, variables related to knowledge on COVID-19 (science-based knowledge and procedural knowledge). The *R*^2^, *F* for change in *R*^2^, and Δ*R*^2^ were calculated for each of the models. Correlation, variance inflation factor (VIF), and tolerance were the diagnostics parameters for multicollinearity. All the correlations between the variables were below 0.6, the VIF range was well below 6.0, and the tolerance was greater than 10 for all variables in the model. The diagnosis dimensions chosen suggested that each of the independent variables has its own contribution to the dependent variable (anxiety symptoms).

Data were analyzed using SPSS (version 25, IBM Corporation, Armonk, NY, USA).

## 3. Results

A total of 561 subjects participated in the study. These included 162 registered nurses, 135 sophomore nursing students, and 264 individuals from the general public. All 162 registered nurses who participated in Part 1 of the study also participated in Part 2 of the study.

Part 1. As shown in the summary of the research participants’ demographic characteristics, nursing students were the youngest, with fewer education years and fewer reported health issues than the nurses and general public groups (Table 1). In addition, an ANOVA showed that nurses’ mean level of education was significantly higher than that of the students and general public, with 69% having at least a Bachelor’s degree. Finally, there were differences between the research groups’ income level, with nurses having a higher income level than students and the general public.

As Table 2 demonstrates, there were significant research group differences in the experienced anxiety levels induced by COVID-19. An ANCOVA controlling for age and education levels and using Bonferroni post hoc tests revealed that nurses experienced significantly higher levels of anxiety than nursing students and the general public (*F*_(2, 258)_ = 3.12, *p* = 0.009; registered nurses > nursing students; registered nurses > general public) with a small to medium effect size (η^2^ = 0.03). No significant difference in anxiety levels was found between the nursing students and the general public. Furthermore, among the nursing cohort, 65% (*n* = 105) reported an anxiety score of >40, suggesting clinically significant symptoms of anxiety, while 26% (*n* = 42) reported an anxiety score of >55, suggesting severe anxiety.

Part 2. Following the finding that the prevalence of COVID-19-induced anxiety among nurses was significantly higher than in the other research groups in Part 1, we conducted Part 2 of the study to focus on the nurse cohort only in order to highlight factors that might impact their anxiety levels. The following variables were tested: nurses’ work settings, i.e., hospital departments or community clinics; their knowledge levels of COVID-19; and their professional seniority, education level, and income level.

Professional seniority had a mean of 16.4 (±12.6) years. Fifty-eight percent of nurses reported working in a hospital setting, while 42% reported working in community clinics (Appendix A). Four nurses (3%) reported working in COVID-19 wards. An ANOVA using Bonferroni post hoc tests revealed that nurses in internal care and geriatric departments experienced significantly higher levels of anxiety compared to other departments (*F*_(4, 158)_ = 2.48, *p* = 0.04), with a medium effect size (η^2^ = 0.06) (Table 3).

The nurses’ procedural knowledge was significantly higher than their science-based knowledge of COVID-19 (66 ± 33, 37 ± 25, respectively, t = 8.86, *p* < 0.001). To further understand the impact of the nurses’ sociodemographic characteristics and their COVID-19 knowledge on their levels of experienced anxiety induced by this pandemic, a hierarchical multiple regression analysis was performed (Table 4). In Model 1, the contribution of the nurses’ sociodemographic characteristics to predicting their anxiety levels was not significant, *F*_(3, 145)_ = 1.23, *p* = 0.29. However, in Model 2, lower professional seniority and working in internal care or geriatric departments significantly accounted for 10% of the variance in anxiety levels, *F*_(2, 143)_ = 7.87, *p* < 0.01. Finally, the addition of the nurses’ science-based knowledge in Model 3 explained an additional significant 4% of the variance in anxiety levels, and the entire Model 3 explained 16% of the variance in anxiety levels, *F*_(2, 141)_ = 3.30, *p* < 0.05.

## 4. Discussion

The current study investigated the impact of the COVID-19-pandemic-induced state of anxiety on three different population groups. This study adds to the existing body of research by evaluating the prevalence of anxiety among the following three groups: nurses, as frontline responders to the COVID-19 pandemic; nursing students experiencing the COVID-19 crisis during their transitional phase to adulthood; and the general public. Overall, our findings show that 58% of all the research participants experienced clinically significant anxiety levels, which is higher than the reported general prevalence of state anxiety symptoms before the COVID-19 pandemic [43]. Studies conducted on the psychological impact of COVID-19 have shown similar results, highlighting this pandemic’s heavy psychological burden [18,20].

Notably, the findings suggest that the prevalence of anxiety was significantly higher among nurses compared to nursing students and the general public. The majority of nurses in our study (65%) experienced clinical symptoms of state anxiety, and 26% reported severe symptoms. This study has a significant contribution to the ongoing debate regarding the psychological impact of the COVID-19 pandemic on nurses compared to other population groups [18,44]. Studies on the psychological impact of COVID-19 on healthcare workers show that nursing staff exhibit a higher prevalence of anxiety than other healthcare workers [20]. However, findings on the prevalence of anxiety in the general public compared to that of nurses are inconsistent [18,44], and before the current study, there were no studies comparing students’ anxiety to that of nurses’ (to the best of our knowledge). The current study suggests that the nurses who participated in the study are more susceptible to experiencing anxiety than both the general public and students. Accordingly, immediate priority should be given to supporting nurses in optimizing effective coping strategies to mitigate their symptoms of anxiety [8].

The highest prevalence of psychological symptoms among nurses in the current study might be attributed to the fact that nurses face a greater risk of exposure to COVID-19 patients as they spend more time on wards. Moreover, this study is the first to show that nurses working in internal care and geriatric departments, which are characterized by older patients who are at a high risk of developing COVID-19 complications [45], reported significantly higher levels of anxiety compared to nurses working in other wards and clinics. This finding suggests that nurses who are in closer contact with older patients might be more exposed to death and moral and ethical concerns, which in turn triggers their anxiety. Interestingly, while age and level of education did not have a significant impact on nurses’ levels of experienced anxiety, seniority in the nursing field showed a significant impact on anxiety levels. Namely, increased anxiety levels were related to less experience in the profession. Less experienced nurses might be concerned about how to deal with new infectious disease nursing and medical guidelines and about quickly learning new skills and applying them in practice. This finding is corroborated by previous studies showing that fewer years of professional experience is related to lower self-efficacy and, as such, to higher levels of psychological distress [46,47].

Notably, the findings suggest that nurses’ science-based knowledge of COVID-19 was negatively associated with anxiety levels, suggesting that an evidence-based understanding of the mechanisms of this novel disease might serve as a protective factor. Research in cognitive psychology indicates that science-based knowledge is needed for structuring causal explanations to better understand how and when to apply certain behaviors [34,48]. Previous research shows that causal links between why and how something works enable people to make predictions, understand implications, draw inferences, and offer explanations that are necessary for problem solving, clinical reasoning, and self-management in acute and chronic illness [32,49,50,51]. Therefore, it follows that accurate knowledge of the causes, consequences, and prevention methods of an infectious disease is an essential condition for engaging in appropriate protective behaviors during an outbreak [35,36]. Consequently, our findings suggest that training sessions for nurses offering a science-based emphasis on COVID-19′s mechanisms are likely to positively impact nurses’ clinical reasoning and understanding of the new protocols and in turn increase their confidence in carrying out their responsibilities and as a result reduce anxiety. While the current study found that seniority and science-based knowledge are protective factors against anxiety among nurses, there may be other factors that protect one from anxiety. For example, exercising was found to be a protective variable against symptoms of anxiety and depression during the COVID-19 pandemic [52]; higher health literacy was associated with lower anxiety symptoms among healthcare workers [53], and low social capital was associated with higher psychological distress during COVID-19 lockdowns [54]. Moreover, satisfaction with teamwork during COVID-19 was associated with low anxiety among healthcare professionals [9]. These variables are beyond the scope of the current study.

Finally, while there is some evidence that student status is a significant risk for developing anxiety [2], our findings show that the anxiety prevalence among our students was similar to that of the general population. This finding can be explained by the mandatory national decision to remove all students from clinical practicums, reducing students’ safety concerns of being exposed to the novel virus to the risk levels of the general population.

This study has several limitations. We conducted a cross-sectional study; thus, this line of research can be extended by incorporating a longitudinal approach to capture the long-term implications of COVID-19-induced anxiety on nurses’ mental wellbeing. In addition, further studies should focus particularly on nurses working in geriatrics and internal medicine wards with the aim of identifying the effects of state anxiety on quality of care, burnout and resilience, illness, and leaving the profession, which have consequences both in the COVID-19 pandemic and for the nursing profession that is suffering from a chronic shortage of nurses worldwide.

Next, the current study assumes that the STAI-S questionnaire is valid for measuring state anxiety that may be induced by the COVID-19 pandemic. This suggests that further studies should incorporate an experimental design to establish the validity of the STAI-S for detecting anxiety induced by a pandemic among the public, students, and healthcare personnel. Further studies should examine both state and trait anxiety and other variables that may be associated with the psychological consequences of the COVID-19 pandemic, such as preexisting health conditions. We recommend using validated measures for wellbeing, such as the PHQ-4, which addresses anxiety and depression symptoms [55], and the Short Form-8 health survey questionnaire for measuring health-related quality of life [56]. Finally, the study was based on a convenience sample, therefore further research is needed with a representative sample and the nursing student group should be compared with a general students group with no health background.

## 5. Conclusions

Nurses are a vulnerable and high-risk population for anxiety symptoms induced by the COVID-19 pandemic. A higher level of deep science-based knowledge of COVID-19 may protect from anxiety. Nursing managers and educators should promote ongoing educational programs for nurses on COVID-19 transmission modes, causes, and possible treatment approaches. This may lower anxiety symptoms among nurses and support them as professionals who are at the forefront of handling the COVID-19 pandemic and its consequences. At the same time, good teamwork, job commitment and dedication, emotional support, and feeling appreciated at work may have the effect of reducing anxiety among healthcare professionals”.

The current findings may be generalized to a possible future national or global emergency disaster. To the extent that there is evidence-based knowledge, this knowledge should be distributed appropriately among the nurses who stand at the forefront of the health battle. This deep science-based knowledge has the potential to mentally boost the nurses and protect them against the consequences of the events for their mental state. Further research is needed to examine the long-term psychological implications of the COVID-19 pandemic for nurses and other healthcare professionals in a variety of work units.

## Figures and Tables

**Table 1 ijerph-19-07070-t001:** Comparison of the sociodemographic characteristics of nurses, nursing students, and the general public (*n* = 561).

Variables	Nurses (*n =* 162)	Nursing Students (*n =* 135)	General Public (*n =* 264)		
	M (SD)	M (SD)	M (SD)	*p*	*F*
Age (years) ^†^	40.7 )11.4)	22.8 (2.8)	40.3 (14.8)	<0.0001	86.7
Education (years) ^†^	16.6 (2.8)	13.5 (0.7)	14.5 (3.0)	<0.0001	54.08
	** *n* ** **(%)**		** *n* ** **(%)**	** *p* **	**χ^2^**
Gender ^‡^				<0.0001	102.7
Female	148 (91)	129 (88)	133 (50)		
Male	15 (9)	16 (12)	131 (50)		
Health status ^‡^				<0.0001	19.40
No health issues	123 (76)	125 (92)	196 (74)		
Health problems	39 (24)	10 (8)	68 (26)		
Family income				=0.003	16.14
Above average	51 (31)	33 (24)	43 (16)		
Average	57 (35)	51 (38)	92 (35)		
Less than average	54 (33)	51 (38)	129 (49)		
Professional level §					
Registered nurses	162 (100)				
Bachelor degree	112 (69.1)				

^†^ ANOVA for continuous variables; ^‡^ chi-square for categorical variables; ^§^ for nurses only.

**Table 2 ijerph-19-07070-t002:** Proportion of registered nurses with clinically significant and severe anxiety levels; results of ANCOVA and Bonferroni post hoc tests on anxiety levels (*n* = 561).

						Anxiety Levels Score	
						Anxiety Score > 40 ^†^	Anxiety Score > 55 ^‡^
	M (SD)	*F*	*p*	η^2^	Bonferroni Post Hoc Tests	*n* (%)	*n* (%)
Registered nurses (*n* = 162)	46 (13.0)	3.12	0.009	0.03	Registered nurses > nursing students Registered nurses > general public	105 (65)	42 (26)
Nursing students (*n* = 135)	42 (13.0)					76 (56)	22 (16)
General public (*n* = 264)	43 (14.0)					145 (55)	50 (19)
Total participants (*n* = 561)	44 (13.5)					325 (58)	112 (20)

**^†^** cut-off score denoting clinically significant anxiety; ^‡^ cut-off score denoting severe anxiety.

**Table 3 ijerph-19-07070-t003:** Analysis of variance (ANOVA) of anxiety levels according to the nurses’ department or clinic of work (*n* = 162).

	M (SD)	*F*	*p*	η^2^	Bonferroni Post Hoc Tests
Community services or outpatient clinics (*n* = 56)	45 (14.2)	2.48	0.04	0.06	Internal medicine or gerontology departments > Community services or outpatient clinics Internal medicine or gerontology departments > Neonatal care, delivery, and pediatrics Internal medicine or gerontology departments > Intensive care units, emergency departmentsInternal medicine or gerontology departments > Others
Internal medicine or geriatric departments (*n* = 28)	53 (10.8)				
Neonatal care, delivery and pediatrics (*n* = 24)	46 (13.9)				
Intensive care units, emergency departments (*n* = 23)	45 (11.2)				
Others (*n* = 30)	43 (12.8)				
Total participants (*n* = 162)	46 (13.2)				

**Table 4 ijerph-19-07070-t004:** Findings from a hierarchical regression analysis for variables accounted for nurses’ COVID-19-induced anxiety levels (*n* = 162).

Variable	Model 1	Model 2	Model 3
	*β*	*Β*	*β*
Age	−0.11	0.35	0.33
Education	−0.08	−0.07	−0.07
Family income level	−0.05	−0.06	−0.08
Seniority in the nursing field		−0.47 *	−0.45 *
Department (internal and geriatric/other departments)		−0.27 ***	−0.26
Science-based knowledge			−0.19 *
Procedural knowledge			0.07
*R* ^2^	0.02	0.12	0.16
*F* for change in *R*^2^	1.23	7.87 **	3.30 *
Δ*R*^2^	0.02	0.10	0.04

* *p* < 0.05; ** *p* < 0.01; *** *p* < 0.001.

## Data Availability

The data presented in this study are available on request from the corresponding author.

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
