# Peer review of "The Role of Science-Based Knowledge on the SARS-CoV-2 Virus in Reducing COVID-19-Induced Anxiety among Nurses"

_ijerph, 2022, doi:10.3390/ijerph19127070_

Round 1

Reviewer 1 Report

This is a cross-sectional study that examined the role of science-based vs. factual/ procedure knowledge on state anxiety among registered nurses during the COVID-19 pandemic in Israel. The study was generally well -presented and the paper was generally well written.  However, several important conceptual and methodological aspects of the study require clarifications and/or modifications.  Below are my comments for the authors' consideration. 

OVERALL COMMENTS:

-Conceptually, the authors claimed the COVID-19 pandemic as a source of stress and state anxiety in the study populations (registered nurses, nursing students, and the general public). Methodologically, the authors claimed that the assessment of state anxiety using STAI-S as an adequate measure of covid-19 induced anxiety. Since these claims were not empirically tested in this study, it is unclear if they were valid. The authors may consider discussing these two points as limitations of the current study and as directions for future research.

-If the study did not only focus on the nurses but the nursing students and the general population as well, part 2 of the study should not be limited to nursing students. On the other hand, if the study did only intend to focus on nurses, it would be important for the authors to avoid over-stating the scope of the study. For instance, page 6, line 214-215 in the Discussion, “The current study investigated the psychological impact of the COVID-19 pandemic 215 on different population groups, in particular nurses.”. 

SPECIFIC COMMENTS BY SECTION:

INTRODUCTION

- The aim of the study was unclear. The authors stated two slightly different aims as follows. Please clarify which one was correct.

        P.2 line 63-64: "This study aims to evaluate the prevalence of psychological distress among nurses compared to other populations."

        P.2 line 77-78: "This study aims to investigate the importance of knowledge, and specifically, the impact of factual-procedural and deep science-based information, on anxiety levels."

-Page 1, lines 31-26: Can the authors explain why they chose to focus on studying state anxiety but not trait anxiety in this study, given that both have been shown to be independently important in Spielberger et al’s work? Specifically, is there evidence in the literature that suggests only the former but not the latter affect the well-being of the nurses or other health professionals in the Covid-19 pandemic?

METHODS

- P.2, line 91-93:  Important information about recruitment was missing.  First, it is unclear how and where the registered nurses were recruited. Please provide the information and comment on the extent to which this sample of registered nurses is representative of all registered nurses in Israel. Second, it is unclear if the university from which the nursing students were recruited was representative of the universities in Israel. Please provide the data and comment. Third, it is unclear if the nursing students from the 2nd year cohort were representative of all nursing students from the selected university. Please provide data and comment.

RESULTS

- P.5-6, Parts 1 and 2, Tables 2 and 3:  Only results of the overall F tests were presented in both sets of analyses. The Bonferroni post hoc tests were mentioned; however, the results were missing. Please present them in the text and/or the tables.

P.6, line 209-210 and Table 4:  The final Model 3 explained only 16% of the total variance of state anxiety. What may be some factors that accounted for the remaining 84%? The authors should discuss this in the Discussion section.

DISCUSSION

- P.6, line 227, “The majority of 226 nurses in our study (65%) experienced clinical symptoms of trait anxiety”: It should be “state”, not “trait” anxiety.

- Page 7, line 280-282: In the authors’ discussion of the limitations, they mentioned “the need for longitudinal studies on anxiety levels that focus on the long-term implications of the COVID-19 crisis on nurses’ mental wellbeing”. Can they elaborate in these studies, how they would be able to tease apart of the effect of nurses’ baseline level of trait anxiety (assuming 2 timepoints) vs. the effect of the Covid-19 pandemic as a stressor? Also, what would be the measures of mental well-being— can they provide some examples? It would be important to have clear answers for these questions, so that the predictors, processes, and outcomes in their overall conceptual model are independent and clear. 

-Page 7, line 282-283: The authors suggested “further studies should focus 282 particularly on nurses working in designated COVID-19 wards.” Please provide the rationale.

MINOR COMMENTS:

- References: The references were numbered in the Reference List at the end; however, a different style (with author names and year of the publication) was used in the main text. Please follow the referencing style of the journal as required.

- Page 8: IRB and Data Availability Statements were missing.

Author Response

Reviewer # 1

Comment 1:

Conceptually, the authors claimed the COVID-19 pandemic as a source of stress and state anxiety in the study populations (registered nurses, nursing students, and the general public). Methodologically, the authors claimed that the assessment of state anxiety using STAI-S as an adequate measure of covid-19 induced anxiety. Since these claims were not empirically tested in this study, it is unclear if they were valid. The authors may consider discussing these two points as limitations of the current study and as directions for future research.

Response:

Thank you for the comment. The assumptions were made based on previous literature which also incorporated the STAI questionnaire (for example, Fernandez et al., 2021; García-González et al., 2021; Tanifuji, et al., 2022).  We elaborate this in the introduction section as follows:

Lines 55-57:

“Spielberger also developed the State-Trait Anxiety Inventory (STAI) to assess anxiety [10], which has been validated among students, adults, medical patients, health care workers, and recently in the context of the COVID-19 pandemic [15]–[17].”

Since this study is a cross-sectional study, we added this limitation, as suggested, to the limitation section:

Lines 345-349:

“Next, the current study assumes the STAI-S questionnaire is valid for measuring state anxiety that may be induced by the COVID-19 pandemic. This suggests that further studies should incorporate an experimental design to establish the validity of the STAIS for detecting anxiety induced by a pandemic among the public, students, and health-care personnel.”

References:

Fernandez, R., Lord, H., Moxham, L., Middleton, R., & Halcomb, E. (2021). Anxiety among Australian nurses during COVID-19. Collegian, 28(4), 357–358. https://doi.org/10.1016/j.colegn.2021.05.002

Tanifuji, T., Aoyama, S., Shinko, Y., Mouri, K., Kim, S., Satomi‐Kobayashi, S., Shinohara, M., Kawano, S., & Sora, I. (2022). Psychological symptoms and related risk factors among healthcare workers and medical students during the early phase of the COVID‐19 pandemic in Japan. Psychiatry and Clinical Neurosciences Reports, 1(1), e5. https://doi.org/10.1002/pcn5.5

García-González, J., Ruqiong, W., Alarcon-Rodriguez, R., Requena-Mullor, M., Ding, C., & Ventura-Miranda, M. I. (2021). Analysis of anxiety levels of nursing students because of e-learning during the Covid-19 pandemic. Healthcare (Switzerland), 9(3), 252. https://doi.org/10.3390/healthcare9030252

Comment 2:

If the study did not only focus on the nurses but the nursing students and the general population as well, part 2 of the study should not be limited to nursing students. On the other hand, if the study did only intend to focus on nurses, it would be important for the authors to avoid over-stating the scope of the study. For instance, page 6, line 214-215 in the Discussion, “The current study investigated the psychological impact of the COVID-19 pandemic 215 on different population groups, in particular nurses.”. 

Response:

Thank you. We rephrased the study’s aims as suggested. Part 1 was conducted to evaluate the prevalence of anxiety distress among nurses compared to other population groups. This was based on previous studies that demonstrated inconsistent results (lines 62-69). Building upon the first part of the study we conducted Part 2, which focused on registered nurses as a group that demonstrated the highest levels of anxiety. Here we were looking for protective factors such as the role of factual-procedural and deep science-based information. We believe that after correcting the research objectives (see below) the issue will be clearer.

The sentence in lines 253-254 was corrected as follows:

“The current study investigated the impact of the COVID-19 pandemic-induced state of anxiety on three different population groups.”

Comment 3:

INTRODUCTION

The aim of the study was unclear. The authors stated two slightly different aims as follows. Please clarify which one was correct.

        P.2 line 63-64: "This study aims to evaluate the prevalence of psychological distress among nurses compared to other populations."

        P.2 line 77-78: "This study aims to investigate the importance of knowledge, and specifically, the impact of factual-procedural and deep science-based information, on anxiety levels."

Response:

Thank you for the comment. We rephrased the aims as follows:

Lines 93-97:

“The research aim is twofold: First, to evaluate the prevalence of state anxiety distress among nurses compared to other groups of populations. Second, to investigate the association between factual-procedural and deep science-based information, and anxiety levels among professional registered nurses. In order to fulfill these aims a two-part study was conducted”.

Comment 4:

Page 1, lines 31-26: Can the authors explain why they chose to focus on studying state anxiety but not trait anxiety in this study, given that both have been shown to be independently important in Spielberger et al’s work? Specifically, is there evidence in the literature that suggests only the former but not the latter affect the well-being of the nurses or other health professionals in the Covid-19 pandemic?

Response:

Indeed, most of the studies dealing with anxiety and COVID-19 refer to “anxiety disorders” with no distinction between “state” anxiety and “trait” anxiety (for example, Global prevalence and burden of depressive and anxiety disorders in 204 countries and territories in 2020 due to the COVID-19 pandemic (thelancet.com). We chose to deal with state anxiety based on the definitions of Harrison et al., 2015. Harrison et al. (2015) summarized that “The trait anxiety would be a life-long expression of worry, as well as a constant stressful response to most situations…. When anxiety occurs in the context of shorter durations (minutes or hours) or in response to a specific threat, and dissipates as the threat weakens, it is considered to be state anxiety.” (Pages 128-129). As we reason throughout the MS, the COVID-19 pandemic like any other crisis most probably induces a temporary emotional state (i.e., state anxiety). Similarly, also other recent studies in the context of COVID-19 incorporated only a state anxiety inventory for public or specific groups (for example, https://pubmed.ncbi.nlm.nih.gov/34314271/).

We clarify this in the introduction and limitation sections, as follows:

Line 53-54:

“State anxiety refers to a specific threat of shorter duration that may disappear as the threat weakens [13]; in this case, the COVID-19 outbreak [14]”.

Line 337-339:

“Further studies should examine both state and trait anxiety and other variables that may be associated with thepsychology consequences of the COVID-19 pandemic.”

Reference:

Peleg, S., Nudelman, G., & Shiloh, S. (2022). COVID-19 state anxiety of older adults: Effects of defensive information processes. Anxiety, Stress and Coping, 35(1), 111–123. https://doi.org/10.1080/10615806.2021.1956479

Comment 5:

METHODS

P.2, line 91-93:  Important information about recruitment was missing.  First, it is unclear how and where the registered nurses were recruited. Please provide the information and comment on the extent to which this sample of registered nurses is representative of all registered nurses in Israel. Second, it is unclear if the university from which the nursing students were recruited was representative of the universities in Israel. Please provide the data and comment. Third, it is unclear if the nursing students from the 2nd year cohort were representative of all nursing students from the selected university. Please provide data and comment.

Response:

Thank you. The nurses were recruited by the polling internet service (line 115-120). It is a convenience sample although the nurse participants were recruited from all over the country and work in a variety of hospitals and wards. A sentence that clarifies this was added on lines 120-121, as follows:

“Although the participants were recruited from all over the country, the sampling method was based on a conveniencesample”.

We also clarify this in the limitations section, lines 337-339:

“Finally, the study was based on a convenience sample, therefore further research is needed with a representative sample”.

The university from which the nursing students were recruited is not necessarily representative of all universities in Israel. Nevertheless, the Department of Nursing from which students were recruited is the largest nursing department in the country, with students from all regions of the country, Jews and Arabs, secular and religious. Also, the sampling method utilized among the students was based on a convenience sample. A sentence that clarifies this was added in lines 112-114:

“The students studying in the university’s Department of Nursing come from all regions of the country, Jews and Arabs, secular and religious. The sampling method was based on a convenience sample”.

Accordingly, this was also outlined in the limitations section, lines 337-339:

“Finally, the study was based on a convenience sample, therefore further research is needed with a representative sample”.

Comment 6:

RESULTS

P.5-6, Parts 1 and 2, Tables 2 and 3:  Only results of the overall F tests were presented in both sets of analyses. The Bonferroni post hoc tests were mentioned; however, the results were missing. Please present them in the text and/or the tables.

Response:

Thank you for the comment. In part 1, line 174, “Bonferroni post hoc tests showed that” was removed from the sentence because it appeared because of a typo.  Now the sentence in lines 187-188 is as follows:

“….an ANOVA showed that nurses’ mean level of education….”

In contrast, the Bonferroni post hoc tests are relevant to Table 2. Therefore, the findings of the Bonferroni test were added in line 193 as follows:

“…registered nurses > nursing students; registered nurses > general public”

The findings are shown in Table 2.

Comment 7:

P.6, line 209-210 and Table 4:  The final Model 3 explained only 16% of the total variance of state anxiety. What may be some factors that accounted for the remaining 84%? The authors should discuss this in the Discussion section.

Response:

A paragraph was added to the discussion section, dealing with other variables that may impact anxiety during the COVID-19 pandemic, lines 292-302, as follows:

“While the current study found that seniority and science-based knowledge are protective factors against anxiety among nurses, there may be other factors that protect one from anxiety. For example, exercising was found to be a protective variable against symptoms of anxiety and depression during the COVID-19 pandemic [52]; higher health literacy was associated with lower anxiety symptoms among healthcare workers [53]; and low social capital was associated with higher psychological distress during COVID-19 lockdowns [54]. Moreover, satisfaction with teamwork during COVID-19 was associated with low anxiety among healthcare professionals [9]. These variables are beyond the scope of the current study”.

In addition, we added the following clarification in the limitation section, lines 322-324:

“Further studies should examine both state and trait anxiety and other variables that may be associated with the psychology consequences of the COVID-19 pandemic.”

References:

Amit Aharon, A., Dubovi, I., & Ruban, A. (2021). Differences in mental health and health-related quality of life between the Israeli and Italian population during a COVID-19 quarantine. Quality of Life Research, 30(6), 1675–1684. https://doi.org/10.1007/s11136-020-02746-5

Caballero-Domínguez, C. C., De Luque-Salcedo, J. G., & Campo-Arias, A. (2021). Social capital and psychological distress during Colombian coronavirus disease lockdown. Journal of Community Psychology, 49(2), 691–702. https://doi.org/10.1002/jcop.22487

Tran, T. V., Nguyen, H. C., Pham, L. V., Nguyen, M. H., Nguyen, H. C., Ha, T. H., Phan, D. T., Dao, H. K., Nguyen, P. B., Trinh, M. V., Do, T. V., Nguyen, H. Q., Nguyen, T. T. P., Nguyen, N. P. T., Tran, C. Q., Tran, K. V., Duong, T. T., Pham, H. X., Nguyen, L. V., … Duong, T. Van. (2020). Impacts and interactions of COVID-19 response involvement, health-related behaviours, health literacy on anxiety, depression and health-related quality of life among healthcare workers: A cross-sectional study. BMJ Open, 10(12). https://doi.org/10.1136/bmjopen-2020-041394

Comment 8:

DISCUSSION

P.6, line 227, “The majority of 269nurses in our study (65%) experienced clinical symptoms of trait anxiety”: It should be “state”, not “trait” anxiety.

Response:

Thank you. correction of the typo was made in line 249.

Comment 9:

Page 7, line 280-282: In the authors’ discussion of the limitations, they mentioned “the need for longitudinal studies on anxiety levels that focus on the long-term implications of the COVID-19 crisis on nurses’ mental wellbeing”. Can they elaborate in these studies, how they would be able to tease apart of the effect of nurses’ baseline level of trait anxiety (assuming 2 timepoints) vs. the effect of the Covid-19 pandemic as a stressor? Also, what would be the measures of mental well-being— can they provide some examples? It would be important to have clear answers for these questions, so that the predictors, processes, and outcomes in their overall conceptual model are independent and clear. 

Response:

Thank you. The limitations section was expanded as suggested (lines 310-326):

“This study has several limitations. We conducted a cross-sectional study; thus, this line of research can be extended by incorporating a longitudinal approach to capture the long-term implications of COVID-19 induced anxiety on nurses’ mental wellbeing. In addition, further studies should focus particularly on nurses working in geriatrics and internal medicine wards in the aim of identifying the effects of state anxiety on quality of care, burnout and resilience, illness, and leaving the profession, which have consequences both for the COVID-19 pandemic and for the nursing profession that is suffering from a chronic shortage of nurses worldwide…. Further studies should examine both state and trait anxiety and other variables that may be associated with the psychological consequences of the COVID-19 pandemic… We recommend using validated measures for well-being, such as the PHQ-4 that addresses anxiety and depression symptoms [56] and the short form-8 health survey questionnaire for measuring health-related quality of life [57]. ‘”

Comment 10:

Page 7, line 282-283: The authors suggested “further studies should focus 282 particularly on nurses working in designated COVID-19 wards.” Please provide the rationale.

Response:

Thank you.  Please see our response to comment #9. The limitations section was elaborated as follows.

Lines 314-317:

“…in the aim of identifying the effects of state anxiety on quality of care, burnout and resilience, illness, and leaving the profession, which have consequences both for the COVID-19 pandemic and for the nursing profession that is suffering from a chronic shortage of nurses worldwide”.

 Comment 11:

MINOR COMMENTS:

References: The references were numbered in the Reference List at the end; however, a different style (with author names and year of the publication) was used in the main text. Please follow the referencing style of the journal as required.

Response:

The references were adapted to the journal’s style.

 Comment 12:

Page 8: IRB and Data Availability Statements were missing.

Response:

The information was added at the end of the article.  Lines 360-361, 362-363.

Institutional Review Board Statement:

The study was approved by the Ethics Committee for research at the university (#1444-1).

Data Availability Statement:

The data presented in this study are available on request from the corresponding author.

Reviewer 2 Report

The title states COVID19 virus, yet SARS-CoV-2 would be the actual virus, i would make the change

Lines 52-53 why would nurses behave different… yes you can say they have knowledge and know how to handle themselves, yet they face patients everyday… Authors should discuss with respect to COVID stress scales by Taylor et al., and ACSS by Delgado-Gallegos this would add a benefit to anxiety and stress

The introduction really does not frame the environment of the nursing staff, give a general vie won anxiety but the title seems disconnected

Methods does not state the criteria for rejection… were underage student or lay people recruited?

Thank you for stating the Cronbach Alpha, I was about to ask on it ?

Authors should include as a supplement the used questionnaire,

Why use hierarchical regression and not a stepwise regression, to remove any potential user bias

Results, authors used 162 registered nurses, is this representative? Particularly would the results in the defined population be able to be generalized… no information of origin r distribution is given as to how the nursing population was obtained, where they from a particular region or countrywide distributed.. same would go for the students and lay people would these populations be representative?

Health problems seems a very broad category do you have more specifics like CVD´s, smokers, inflammation issues, as this brings about particular fears of the individual

Is Family income related to national per capital earnings? Or just a general mid point, if so does it mean anything… do nurses get Good or bad salaries in Israel with reference to the general population, would be better to distribute by a particular number in earning say $10,000 more or less or some value appropriate for the country

58% hospital and 42% community clinics, yet only 3 nurses claimed to be in COVID wards, seems bit small granted not fully familiar with Israel COVID incidence through, so I will not make emphasis, yet Im missing supplemental material 1

Overall, our findings show that 58% of all the research participants experienced clinically 220 significant anxiety levels, which is higher than the reported general prevalence of state anxiety symptoms before the COVID-19 pandemic (Bradley, 2016).   How was the population determined and is it representative, I again go back to this term…

The population sampling also becomes an issue as authors also state.. However, findings on the prevalence of anxiety in the general public compared to that of nurses are inconsistent… well is it regional?

Similar here… The current study highlights that nurses are more susceptible to experiencing anxiety than both the general public and students

I believe to an extent the authors are overgeneralizing, now this generalization may be correct but as before please add the sampling and the representativity

Author Response

Reviewer # 2

Comment 1:

The title states COVID19 virus, yet SARS-CoV-2 would be the actual virus, i would make the change

Response:

The title name has been changed as follows:

The role of science-based knowledge on the SARS-CoV-2 virus in reducing COVID-19 induced anxiety among nurses”.

Comment 2:

Lines 52-53 why would nurses behave different… yes you can say they have knowledge and know how to handle themselves, yet they face patients everyday… Authors should discuss with respect to COVID stress scales by Taylor et al., and ACSS by Delgado-Gallegos this would add a benefit to anxiety and stress. The introduction really does not frame the environment of the nursing staff, give a general vie won anxiety but the title seems disconnected

Response:

Thank you. We added the references as suggested and we further elaborated why nurses are in a risk group for experiencing anxiety, lines 62-69:

“Working in the frontline involved wearing personal protective equipment and continuous fear of self-infection or infecting someone in their family. Given that this can trigger anxiety, recent studies pointed out that working with COVID patients was associated with traumatic stress and COVID Stress Syndrome [22], [23].”

References:

Delgado-Gallegos, J. L., Montemayor-Garza, R. de J., Padilla-Rivas, G. R., Franco-Villareal, H., & Islas, J. F. (2020). Prevalence of stress in healthcare professionals during the Covid-19 pandemic in Northeast Mexico: A remote, fast survey evaluation, using an adapted Covid-19 stress scale. International Journal of Environmental Research and Public Health, 17(20), 1–12. https://doi.org/10.3390/ijerph17207624

Taylor, S. (2021). COVID Stress Syndrome: Clinical and Nosological Considerations. In Current Psychiatry Reports (Vol. 23, Issue 4, pp. 1–7). Springer. https://doi.org/10.1007/s11920-021-01226-y

Comment 3:

 Methods does not state the criteria for rejection… were underage student or lay people recruited? Thank you for stating the Cronbach Alpha, I was about to ask on it ?

Response:

Thank you. Three groups of participants were recruited for the study:

  1. Registered nurses
  2. Participants from the general public aged 18+
  3. Nursing students in the second year of their studies (out of 4 years of study), all adults.

To clarify the exclusion criteria we added the following:

Lines 107-108 as follows:

“….and lay adult individuals from the general public who were not working in healthcare settings”

Line 112-114:

“The students studying in the university’s Department of Nursing come from all regions of the country, Jews and Arabs, secular and religious. The sampling method was based on a convenience sample.”.

Comment 3:

Authors should include as a supplement the used questionnaire,

Response:

Thank you. We translated and added several items from the questionnaire, please see Supplemental Material 1.

Comment 4:

Why use hierarchical regression and not a stepwise regression, to remove any potential user bias

Response:

The hierarchical regression model was chosen based on theoretical reasons (Tabachnick & Fidell, 2014). Previous studies found an association between anxiety and socioeconomic variables and working conditions (for example, Pappa et al., 2020). Therefore, we wanted to control the order of the variables inserted in the regression equation: first the SE variables, then the nurses’ working variables, and finally variables related to knowledge regarding COVID-19 (science-based knowledge and procedural knowledge). In that way we were able to assess the odds of each group of variables at its own point of entry. The stepwise regression, i.e., the order of the variables inserted in the equation, is based solely on statistical criteria (computed from the particular sample) with no researcher control. The outcome of the two regression methods could be the same, but in the stepwise method we lose part of the researchers’ interpretation.

Comment 5:

Results, authors used 162 registered nurses, is this representative? Particularly would the results in the defined population be able to be generalized… no information of origin r distribution is given as to how the nursing population was obtained, where they from a particular region or countrywide distributed.. same would go for the students and lay people would these populations be representative?

Response:

The nurses were recruited by an online polling service (lines 115-119). It is a convenient sample although the nurse participants were recruited from all over the country and work in a variety of hospitals and wards.

A sentence that clarifies this was added in lines 120-121:

“Although the participants were recruited from all over the country, the sampling method was based on a convenience sample”.

And it was added to the limitation section, lines 327-328:

“Finally, the study was based on a convenience sample, therefore further research is needed with a representative sample”.

The university from which the nursing students were recruited is not necessarily representative of all universities in Israel. Nevertheless, the students studying in the university’s Department of Nursing, come from all regions of the country, Jews and Arabs, secular and religious. Also, among the students the sampling method was based on a convenience sample.

A sentence that clarifies this was added on lines 112-114:

“The students studying in the university’s Department of Nursing come from all regions of the country, Jews and Arabs, secular and religious. The sampling method was based on a convenience sample”.

Accordingly, it was added to the limitation section, lines 327-328:

“Finally, the study was based on a convenience sample, therefore further research is needed with a representative sample”.

Comment 6:

Health problems seems a very broad category do you have more specifics like CVD´s, smokers, inflammation issues, as this brings about particular fears of the individual

Response:

Thank you. We asked participants about health issues, whether they have such issues or not; however, we didn’t collect specific data regarding this. We elaborate on this within the limitations section as follows:

Lines 322-324:

“Further studies should examine both state and trait anxiety and other variables that may be associated with the psychological consequences of the COVID-19 pandemic, such as preexisting health conditions”

Comment 7:

Is Family income related to national per capital earnings? Or just a general mid point, if so does it mean anything… do nurses get Good or bad salaries in Israel with reference to the general population, would be better to distribute by a particular number in earning say $10,000 more or less or some value appropriate for the country

Response:

The salary of nurses in Israel is average.

We used the question regarding the family average income in accordance with the measurement accepted by the Central Bureau of Statistics in Israel and in accordance with that accepted in studies conducted in Israel. The research significance is that the online polling service used in this study provided sampling according to household income well-being indicators recommended by The National Centre for Social and Economic Modelling (NATSEM), University of Canberra(2017). The current research found that income was not associated with anxiety.

The related item within the questionnaire was phrased as follows:

According to the Central Bureau of Statistics in Israel, the average family income is NIS 17,300. Considering this, is your family income is above/below or equal to the average?

Comment 8:

58% hospital and 42% community clinics, yet only 3 nurses claimed to be in COVID wards, seems bit small granted not fully familiar with Israel COVID incidence through, so I will not make emphasis, yet Im missing supplemental material 1

Response:

Indeed, a limited number of nurses who had worked in the Corona wards participated in the current study. However, the study was conducted at the end of the first wave of corona, with a relatively low number of hospitalized patients and low number of deaths from corona (see lines 28-32). Hospitals preferred to refer intensive care nurses to the Corona wards rather than nurses from internal care and other wards. Thus, at the time of the study, it makes sense that there would be a minority of nurses in the sample who had actually worked in the Corona wards.

As written earlier, the fact that this was a convenience sample of nurses was included in the method, lines 120-121, and in the study limitations, in lines 327-328.

 Comment 9:

Overall, our findings show that 58% of all the research participants experienced clinically 220 significant anxiety levels, which is higher than the reported general prevalence of state anxiety symptoms before the COVID-19 pandemic (Bradley, 2016).   How was the population determined and is it representative, I again go back to this term…

Response:

The issue of the representativeness of the sample was expanded in the methods section as follows:

Line 112-114:

“The students studying in the university’s Department of Nursing come from all regions of the country, Jews and Arabs, secular and religious. The sampling method was based on a convenience sample”.

Lines 120-121:

“Although the participants were recruited was from all over the country, the sampling method was based on a convenience sample”.

And in the limitations section, lines 328-329:

“Finally, the study was based on a convenience sample, therefore further research is needed with a representative sample and the nursing student group should be compared with a general students’ group with no health background”.

Comment 10:

The population sampling also becomes an issue as authors also state.. However, findings on the prevalence of anxiety in the general public compared to that of nurses are inconsistent… well is it regional? Similar here… The current study highlights that nurses are more susceptible to experiencing anxiety than both the general public and students

Response:

Thank you.  Although the study is based on a convenience sample, since we used an online polling service that enables rapid attainment of responses with representative sampling by socioeconomic status, gender, age, and profession from every region in Israel, we assume that the sample may be reliable. Therefore, we suggest that the difference in anxiety levels points to nurses as a high-risk group. This finding was supported by the most recent studies as well.

We elaborate on this as follows, lines 257-258:

“The current study suggests that the nurses who participants in the study are more susceptible to experiencing anxiety than both the general public and students”.

Comment 11:

I believe to an extent the authors are overgeneralizing, now this generalization may be correct but as before please add the sampling and the representativity

Response:

Thank you. We accept the comment and corrections were made throughout the manuscript, particularly regarding the sampling method, limitations section, and the conclusion. All the corrections are highlighted in yellow.

Reviewer 3 Report

INTRODUCTION

The logic thread is weak. There are gaps, which don’t permit to structure a solid and linear thinking.

Line 30: Stress was cited but not discussed

Line 39: Which negative consequences high levels of anxiety have?

Line 39-40: The correlation between the influence of anxiety and of the COVID-19 is insufficient

Line 40: What is the meaning of “higher level of COVID-19” in this sentence?

Line 42-44: It must be discussed the discrepancy between the management and the care of psychophysiological symptoms due to anxiety with and without the COVID-19 pandemic. 

Line 50-51: Authors sustain that all "nurses, as frontline responders to the COVID-19 pandemic" have more anxiety but the cited studies reference to all health professional without a specific focus on the nurses.
The COVID-19 pandemic influences generally all the healthcare sectors. In this case, why wasn't the survey extended also to the other health professional such as these of the rehabilitation, prevention, and diagnostic areas? 

Line 55 and 61-62: I think other vulnerable groups of population, such as elderly, weren’t forgotten. Soo, the framework doesn’t support fully the choosing of the sample.

Line 69-70: Why the patients? The sample is different so this paragraph is completely inopportune.

Line 73: There is only one reference to justify the second part of the research. The topic must be more developed! 

METHOD

Research design and participants: The logical thread is frail without a coherence between the paragraphs. The presentation of this kind of study design must be better justified, while the criteria for the selection of the target group aren’t clear.

Why did you choose just nursing students among all university population?***

How do you say that the general population have not a healthcare background?

Line 87-90: This paragraph could be insert in the introduction as description of the context.

Line 81-83: How many times was during the collection of responses?

Data collection instruments part 2: The used questionnaire isn’t available also in the cited article so it’s difficult to understand the way for assessing the COVID-19 knowledge of respondents. The questions of the questionnaire must be insert.

RESULTS

Ok.

DISCUSSION

Line 274-278: This consideration has a lot of limits such as:

  • The considered students have chosen a health university course, so they are vocated to face up also heavy health issues.
  • There isn’t a comparison with other kind of university students, which is fundamental to deduct the similar anxiety of the student population in comparison with the general population
  • Students can include also attenders of the high school

Other proposals for reducing anxiety in nurses can be presented:

  • Psychological support
  • Organizational changes
  • Co-working and team building

CONCLUSIONS

Line 287-289: As said, these conclusions are limited to the frontal education of the nurses while other instruments, such as audits, co-working and so on, can improve the deep science-based knowledge of COVID-19 in this target

Line 292-293: A focus on the nurses working in geriatrics!

Author Response

Reviewer # 3

Comment 1:

INTRODUCTION

The logic thread is weak. There are gaps, which don’t permit to structure a solid and linear thinking.

Response:

Thank you for the comment. The introduction section was re-organized and expanded. We believe that it is now more logical and clear.

For example, a sentence with references was added in lines 42-43 as follows:

“A high level of pandemic anxiety was associated with higher burnout, depression, and fear among nurses and other healthcare workers [9,10]”

A paragraph was moved from the middle of the introduction to the end of the section, lines 85-88. Now we believe it is more reasonable. Also, the paragraph was expanded as follows (lines 47-55):

“Spielberger et al. differentiated between anxiety as a temporary emotional “state” and as a consistent personality attribute or “trait” [12]. The current study focuses on state anxiety, which refers to a condition of feeling fearful, tense, apprehensive, nervous, and worried due to physiological arousal or a threatening stimulus. State anxiety refers to a specific threat of shorter duration that may disappear as the threat weakens [13]; in this case, the COVID-19 outbreak [14]”.

The aims of the study were re-written at the end of the introduction, lines 91-95 as follows:

“The research aim was twofold: First, to evaluate the prevalence of state anxiety distress among nurses compared to other population groups. Second, to investigate the association between factual-procedural and deep science-based information, and anxiety levels among professional registered nurses. In order to fulfill this aims a two-part study was conducted”.

References:

Pappa, S., Athanasiou, N., Sakkas, N., Patrinos, S., Sakka, E., Barmparessou, Z., Tsikrika, S., Adraktas, A., Pataka, A., Migdalis, I., Gida, S., & Katsaounou, P. (2021). From recession to depression? Prevalence and correlates of depression, anxiety, traumatic stress and burnout in healthcare workers during the Covid-19 pandemic in Greece: Amulti-center, cross-sectional study. International Journal of Environmental Research and Public Health, 18(5), 1–16. https://doi.org/10.3390/ijerph18052390

Teo, I., Chay, J., Cheung, Y. B., Sung, S. C., Tewani, K. G., Yeo, L. F., Yang, G. M., Pan, F. T., Ng, J. Y., Bakar Aloweni, F. A., Ang, H. G., Ayre, T. C., Chai-Lim, C., Chen, R. C., Heng, A. L., Nadarajan, G. D., Hock Ong, M. E., See, B., Soh, C. R., … Tan, H. K. (2021). Healthcare worker stress, anxiety and burnout during the COVID-19 pandemic in Singapore: A 6-month multi-centre prospective study. PLoS ONE, 16(10 October), e0258866. https://doi.org/10.1371/journal.pone.0258866

Comment 2:

Line 30: Stress was cited but not discussed

Response:

The word “stress” was removed from the sentence.

Comment 3:

Line 39: Which negative consequences high levels of anxiety have?

Response:

We elaborate on the negative consequences of anxiety as suggested, lines 42-43:

“A high level of pandemic anxiety was associated with higher burnout, depression, and fear among nurses and other healthcare workers (Pappa et al., 2021; Teo et al., 2021)”.

Comment 4:

Line 39-40: The correlation between the influence of anxiety and of the COVID-19 is insufficient

Response:

Thank you. We elaborate as follows, lines 64-67:

“A systematic review and meta-analysis by Pappa et al. (2020) concluded that the pooled prevalence of anxiety among healthcare workers during the COVID-19 pandemic was 23.21% (95% CI 17.77-29.13, I2=99%), and among nurses and physicians 17.93% suffered from mild anxiety”.

Comment 5:

Line 40: What is the meaning of “higher level of COVID-19” in this sentence?

Response:

The typo was corrected as follows (now in lines 38-39):

“….higher levels of anxiety regarding COVID-19 on psychophysiological health”.

Comment 6:

Line 42-44: It must be discussed the discrepancy between the management and the care of psychophysiological symptoms due to anxiety with and without the COVID-19 pandemic. 

Response:

Thank you. We elaborated the paragraph to make it clear that we are discussing anxiety induced by the pandemic (lines 38-44)

Comment 7:

Line 50-51: Authors sustain that all "nurses, as frontline responders to the COVID-19 pandemic" have more anxiety but the cited studies reference to all health professional without a specific focus on the nurses.
The COVID-19 pandemic influences generally all the healthcare sectors. In this case, why wasn't the survey extended also to the other health professional such as these of the rehabilitation, prevention, and diagnostic areas? 

Response:

Thank you. Indeed, healthcare professionals are exposed to COVID-19 patients and may suffer from a higher level of anxiety compared to laymen. In the MS we clarify why we believe that they might be more susceptible to COVID-19 anxiety: “Studies show that nurses, as frontline responders to the COVID-19 pandemic, have been reported to experience the highest prevalence and levels of anxiety [19-21].” The references are studies that compared the anxiety of nurses and of other healthcare professionals. For example, Pappa et al. (2021), in their systematic review and meta-analysis, found that the prevalence of anxiety among nurses was 25.8%, compared to doctors with 21.73%.

However, there is certainly room to expand the research to more health professions and diverse work units. Following the comment, a sentence was expanded in the authors’ recommendation in the conclusion section, lines 341-343:

“Further research is needed to examine the long-term psychological implications of the COVID-19 pandemic for nurses and other healthcare professionals in a variety of work units”.

Comment 8:

Line 55 and 61-62: I think other vulnerable groups of population, such as elderly, weren’t forgotten. Soo, the framework doesn’t support fully the choosing of the sample.

Response:

We strongly agree that there are other high-risk groups such as the elderly. However, as we noted in response to the previous comment, in the current study we decided to focus on nurses. Of course, the research can be extended to subgroups of the general public, in a similar method as in the current study.

Comment 9:

Line 69-70: Why the patients? The sample is different so this paragraph is completely inopportune.

Response:

Thank you for the comment. The sentence was omitted.

Comment 10:

Line 73: There is only one reference to justify the second part of the research. The topic must be more developed! 

Response:

Thank you. We expanded the justification for the second part of the research as follows (lines 87-90):

“Having more cohesive causal information promotes better retrieval of previously learned facts, allows people to make predictions, understand implications, draw inferences, and offer explanations – all of which are necessary for problem-solving protective behaviors during an infectious disease outbreak [35]–[37]. “

We added the following references:

Russ, R. S., Scherr, R. E., Hammer, D., & Mikeska, J. (2008). Recognizing mechanistic reasoning in student scientific inquiry: A framework for discourse analysis developed from philosophy of science. Science Education, 92(3), 499–525. https://doi.org/10.1002/sce.20264

Dubovi, I., Levy, S. T., Levy, M., Zuckerman Levin, N., & Dagan, E. (2020). Glycemic control in adolescents with type 1 diabetes: Are computerized simulations effective learning tools? Pediatric Diabetes, 21(2), 328–338. https://doi.org/10.1111/pedi.12974

Amit Aharon, A., Ruban, A., & Dubovi, I. (2020). Knowledge and information credibility evaluation strategies regarding COVID-19: A cross-sectional study. Nursing Outlook, 0(0). https://doi.org/10.1016/j.outlook.2020.09.001

Dagan, E., Dubovi, I., Levy, M., Zuckerman Levin, N., & Levy, S. T. (2019). Adherence to diabetes care: Knowledge of biochemical processes has a high impact on glycaemic control among adolescents with type 1 diabetes. Journal of Advanced Nursing, 75(11), 2701–2709. https://doi.org/10.1111/jan.14098

Hastie, R. (2015). Causal thinking in judgments. In The Wiley Blackwell Handbook of Judgment and Decision Making(pp. 590–628). John Wiley & Sons, Ltd. https://doi.org/10.1002/9781118468333.ch21

Todd, A., & Romine, W. (2018). The learning loss effect in genetics: What ideas do students retain or lose after instruction? CBE Life Sciences Education, 17(4). https://doi.org/10.1187/cbe.16-10-0310

Piltch-Loeb, R., Zikmund-Fisher, B. J., Shaffer, V. A., Scherer, L. D., Knaus, M., Fagerlin, A., Abramson, D. M., & Scherer, A. M. (2019). Cross-Sectional Psychological and Demographic Associations of Zika Knowledge and Conspiracy Beliefs Before and After Local Zika Transmission. Risk Analysis, 39(12), 2683–2693. https://doi.org/10.1111/risa.13369

Taylor, M., Raphael, Barr, Agho, Stevens, & Jorm. (2009). Public health measures during an anticipated influenza pandemic: Factors influencing willingness to comply. Risk Management and Healthcare Policy, 9. https://doi.org/10.2147/rmhp.s4810

Comment 11:

METHOD

Research design and participants: The logical thread is frail without a coherence between the paragraphs. The presentation of this kind of study design must be better justified, while the criteria for the selection of the target group aren’t clear.Why did you choose just nursing students among all university population?

Response:

Thank you. The student sample for the current study consisted of second year nursing students. They all had clinical experience according to the study program. Since these were second-year students (of a 4-year program) they were not part of the efforts made to treat Corona patients, neither in the hospitals nor in the community. The authors were able to recruit students only from the nursing department due to the ethics approval limitations, which included limited access during the first lockdown.

Following the reviewer’s kind suggestion, we elaborate on this in the limitations section, lines 323-325, as follows:

“Finally, the study was based on a convenience sample, therefore further research is needed with a representative sampleand the nursing student group should be compared with a general students’ group with no health background”.

Comment 12:

How do you say that the general population have not a healthcare background?

Response:

We did not mean to say that the general population have no healthcare background. We meant that the exclusion criteria among the public group should be people who are not working in healthcare settings. Therefore, the sentence was rephrased as follows, line 105-106:

“….and lay individuals from the general public who were not working in healthcare settings.”

Comment 13:

Line 87-90: This paragraph could be insert in the introduction as description of the context.

Response:

Thank you. We reorganized the introduction section as suggested.

Lines 28-32, as follows:

“In Israel, as of mid-May 2020, there were 16,689 confirmed cases of COVID-19 in Israel, with 266 deaths (a prevalence of 29/1 million citizens). In late April and early May, Israel was ranked 47th in the number of deaths per one million citizens and 25th in the number of confirmed cases per one million citizens relative to 210 countries worldwide [1]”.  

Comment 14:

Line 81-83: How many times was during the collection of responses?

Response:

Thank you. We approached each participant only once.

 Comment 15:

Data collection instruments part 2: The used questionnaire isn’t available also in the cited article so it’s difficult to understand the way for assessing the COVID-19 knowledge of respondents. The questions of the questionnaire must be insert.

Response:

Thank you. We translated and added four items from the questionnaire in Supplemental Material 1.

 Comment 16:

DISCUSSION

Line 274-278: This consideration has a lot of limits such as:

  • The considered students have chosen a health university course, so they are vocated to face up also heavy health issues.
  • There isn’t a comparison with other kind of university students, which is fundamental to deduct the similar anxiety of the student population in comparison with the general population
  • Students can include also attenders of the high school

Response:

We understand the comment. The student sample for the current study consisted of second year nursing students. They all had clinical experience according to the study program. At the time of the study, according to the university guidelines, studies at the university were conducted remotely on the Zoom platform. Nursing students did not attend clinical studies in the hospital wards where they were studying. Since these were second-year students (of a 4-year program), they were not part of the efforts to treat Corona patients, neither in the hospitals nor in the community. In fact, the general population served as a comparison group.

According to the comment we added this issue to the limitations section, lines 322-324, as follows:

“Finally, the study was based on a convenience sample, therefore further research is needed with a representative sampleand the nursing student group should be compared with a general students’ group with no health background”.

Comment 16:

Other proposals for reducing anxiety in nurses can be presented:

  • Psychological support
  • Organizational changes
  • Co-working and team building

Response:

We agree with the comment. Nevertheless, we focused on science-based knowledge and procedural knowledge as the two main explanatory variables associated with a reduction in state anxiety. We did not conduct a comprehensive study with a range of variables that could explain the anxiety among nurses. A paragraph that expanded this issue was added as follows:

Lines 289-297, discussion section:

“While the current study found that seniority and science-based knowledge are protective factors against anxiety among nurses, there may be other factors that protect one from anxiety. For example, exercising was found to be a protective variable against symptoms of anxiety and depression during the COVID-19 pandemic [52]; higher health literacy was associated with lower anxiety symptoms among healthcare workers [53]; and low social capital was associated with higher psychological distress during COVID-19 lockdowns [54]. Moreover, satisfaction with teamwork during COVID-19 was associated with low anxiety among healthcare professionals [9]. These variables are beyond the scope of the current study”.

Comment 17:

CONCLUSIONS

Line 287-289: As said, these conclusions are limited to the frontal education of the nurses while other instruments, such as audits, co-working and so on, can improve the deep science-based knowledge of COVID-19 in this target:

Response:

Reference was made to the issues raised throughout the article in accordance with the comments of the reviewer. Following the comment, sentences were added to the conclusions as follows:

Lines 332-334:

“At the same time, good teamwork, job commitment and dedication, emotional support and feeling appreciated at work may have the effect of on reducing anxiety among healthcare professionals”.

Line 340-342

“Further research is needed to examine the long-term psychological implications of the COVID-19 pandemic for nurses and other healthcare professionals in a variety of work units”.

Comment 18:

Line 292-293: A focus on the nurses working in geriatrics!

Response:

Following the comment, a correction was made in line 308 as follows:

“…particularly on nurses working in geriatrics and internal medicine wards in the aim ….”

Round 2

Reviewer 1 Report

In this revision, the authors have responded to my previous comments satisfactorily.  The manuscript has improved considerably.

Reviewer 2 Report

All details have been answered